# An opioid-like system regulating feeding behavior in *C. elegans*

**Mi Cheong Cheong[1]\*[†], Alexander B Artyukhin[1][‡], Young-Jai You[2], Leon Avery[1]**

[1]Department of Physiology and Biophysics, Virginia Commonwealth University, Richmond, United States; [2]Department of Biochemistry and Molecular Biology, Virginia Commonwealth University, Richmond, United States

**Abstract** Neuropeptides are essential for the regulation of appetite. Here we show that neuropeptides could regulate feeding in mutants that lack neurotransmission from the motor neurons that stimulate feeding muscles. We identified *nlp-24* by an RNAi screen of 115 neuropeptide genes, testing whether they affected growth. NLP-24 peptides have a conserved YGGXX sequence, similar to mammalian opioid neuropeptides. In addition, morphine and naloxone respectively stimulated and inhibited feeding in starved worms, but not in worms lacking NPR-17, which encodes a protein with sequence similarity to opioid receptors. Opioid agonists activated heterologously expressed NPR-17, as did at least one NLP-24 peptide. Worms lacking the ASI neurons, which express *npr-17*, did not response to naloxone. Thus, we suggest that *Caenorhabditis elegans* has an endogenous opioid system that acts through NPR-17, and that opioids regulate feeding via ASI neurons. Together, these results suggest *C. elegans* may be the first genetically tractable invertebrate opioid model.

**\*For correspondence:**
mccheong81@gmail.com

**Present address:** [†]Department of Pharmacology, University of Texas Southwestern Medical Center at Dallas, Dallas, United States; [‡]Boyce Thompson Institute, Cornell University, Ithaca, United States

**Competing interests:** The authors declare that no competing interests exist.

**Reviewing editor**: Oliver Hobert, Columbia University, United States

## Introduction

Animal feeding behavior has been well studied because of its health relevance as well as its fundamental physiological and ecological importance. *Caenorhabditis elegans* is a useful model system for studying the genetic control of feeding (*Avery, 2012*). In *C. elegans*, feeding is dependent on the pumping motion of the pharynx, which results in the ingestion of food, and whose rate is an easily quantified measure of feeding. The pathways that control feeding are at least partly conserved from worms to mammals.

Control of animal feeding is not simple. Multiple peripheral and central signals, neuropeptides among them, modulate food intake. For example, α-melanocyte-stimulating hormone, a peptide which activates MC3 and MC4 melanocortin receptors and inhibits food intake (*Meister, 2007*), putative satiety signal peptide $YY_{3-36}$ ($PYY_{3-36}$) (*Batterham et al., 2002*), and cholecystokinin (*Gibbs et al., 1973*) participate in the termination of meals in mammals. Galanin is a peptide that stimulates feeding in satiated rats after intraventricular or intrahypothalamic injection. Opioids are also involved in controlling food intake, and they have many other biological effects (e.g., analgesia, drug addiction, and immune response). For example, the opioid agonist morphine increases food intake in mammals while the opioid blocker, naloxone, significantly decreases food intake (*Martin et al., 1963*).

The opioid system is composed of μ-opioid receptors (MORs), δ-opioid receptors (DORs), and κ-opioid receptors (KORs), and endogenous ligands for these receptors. Enkephalins, dynorphins, and β-endorphin peptides are produced by proteolytic cleavage of large protein precursors known as preproenkephalin, preprodynorphin, and proopiomelanocortin (POMC), respectively. These peptides form the opioid family. All opioid peptides share a common N-terminal YGGF signature sequence, which interacts with opioid receptors (*Holtzman, 1974*; *Akil et al., 1998*). Many researchers have shown that the opilioid system modulates food intake. β-endorphin stimulates food intake when

**eLife digest** When and how much an animal eats is controlled by a complex web of signals that are produced by the animal's body and brain. Molecules called opioid neuropeptides are among these signals, and act to control eating in mammals by binding to receptors in the brain and body. These receptors can also bind to similar molecules called opiates (such as morphine); opiates are amongst the oldest drugs used by humans and have diverse effects ranging from pain relief to addiction. While the activities of opiates and opioid neuropeptides have been studied in mammals, relatively little is known about opioid signaling in simpler animals.

The mechanisms behind many biological processes have been investigated using a worm called *C. elegans* as a model system because it has a simple body plan and its genes can be altered easily. The feeding behavior of *C. elegans* is no exception. This worm feeds by contracting and relaxing its pharyngeal muscle to move food into its gut. When the worms sense that food is available, this 'pharyngeal pumping' is regulated by one type of nerve cell. Slow pharyngeal pumping also continues in starved worms when food is not available, possibly to encourage them to eat new potential sources of food. However, this slow pumping does not require the same type of nerve cell.

Cheong et al. hypothesized that the slow pumping in starved worms might depend on neuropeptide signaling instead, and have now tested this idea using engineered worms that made lower levels of a number of these molecules. The experiments uncovered a molecule called NLP-24 that promotes the slow pharyngeal pumping. This molecule is similar to opioid neuropeptides found in mammals. Worms that made less NLP-24 than normal grew more slowly; this suggests that they had problems feeding. Moreover, the levels of NLP-24 were found to increase in normal worms soon after they were deprived of food. Further experiments revealed the identity of the receptor for this molecule, which is also similar to mammalian opioid receptors.

The discovery that opioid signaling is involved in *C. elegans'* feeding behavior may well, in future, also help to identify new molecular players involved in opioid signaling. Further studies might also help the search for ways to reduce the problematic side-effects that limit the usefulness of opiate drugs as medicines.

administrated directly into the VMH (ventromedial hypothalamus) (*Grandison and Guidotti, 1977*). Selective agonists for the μ receptor (DAMGO), the δ receptor (DADLE), and the κ receptor (U50448) also increase food intake (*Tepperman and Hirst, 1983*; *Gosnell et al., 1986*; *Jackson and Cooper, 1986*). Furthermore, β-endorphin levels are associated with overeating in genetically obese mice and *fa/fa* rats (*Margules et al., 1978*). There is abundant pharmacological evidence that the opioid system is present and controls feeding in both invertebrates and vertebrates (*Harrison et al., 1994*). A simple invertebrate model system would be a useful starting point for understanding more complex ones. Furthermore, it might shed light on the evolutionary origin of the opioid system. However, no opioid system has yet been molecularly identified in invertebrates.

Peptide hormones and peptide neurotransmitters are well conserved from humans to invertebrates (*Kimura et al., 1997*; *Janssen et al., 2008b*). Among the most conserved are those that signal nutritional state and control feeding and metabolism—insulin is the best-known example. *C. elegans* has 115 neuropeptide genes encoding over 250 distinct neuropeptides (*Li and Kim, 2008*). However, the functions of most neuropeptides remain unknown. Here we show that, like humans, *C. elegans* has opioid peptide signals that regulate feeding. We found that morphine and naloxone, respectively, stimulate and inhibit feeding in starved wild-type worms, but not in worms lacking the G-protein coupled receptor NPR-17. Our genetic screens identified NLP-24-derived peptides as possible endogenous opioid ligands on which feeding motions that occur in the absence of food may depend and NPR-17 as the opioid receptor on which they act. To our knowledge this is the first genetically tractable invertebrate opioid model.

## Results

### Peptide signals control MC-independent pumping

In well-fed *C. elegans*, pharyngeal pumping, the motion of the feeding muscles, is mostly determined by MC motor neuron activity in the pharynx (*Raizen et al., 1995*). The food stimulates the MC neuron,

which fires at the beginning of each pump, triggering a muscle action potential that makes the muscle contract. But a starved worm continually pumps at a low rate even in the absence of food. This pumping does not require MC neurons. This MC-independent pumping makes worms viable without MC, although they pump slowly, grow slowly, and are small and pale. We were curious how worms continue to pump without MC.

In mammals, neuropeptides modulate food intake. Over 250 neuropeptides encoded by 115 genes have been identified in *C. elegans* (*Li and Kim, 2008*). Most of their functions are still unknown. We hypothesized that MC-independent pumping might depend on neuropeptide signaling. To test our hypothesis, we made *eat-2*; *egl-3* mutants, in which signals from both MC and neuropeptides are disrupted. *eat-2* encodes an acetylcholine receptor subunit specific to the MC → muscle synapse; thus the mutants are functionally MC-minus (*McKay et al., 2004*). *egl-3* encodes a protease necessary for neuropeptide processing, and thus the mutant lacks most neuropeptides (*Husson et al., 2006*). *egl-3* mutants pumped at the wild-type rate. However, *eat-2*; *egl-3* mutants pumped only 1/3 as fast as *eat-2* mutants (*Figure 1A*). This result indicated that neuropeptides regulate pumping in the absence of MC neuron activity.

To identify neuropeptides that regulate pumping in the absence of MC function, we performed an RNAi (RNA interference) screen of neuropeptide genes. Because RNAi works less efficiently in the nervous system than in other tissues, we first tested *egl-3* RNAi efficiency in two genetic backgrounds that enhance RNAi in neurons, *rrf-3* and *eri-1*; *lin-15b* (*Simmer et al., 2002*; *Wang et al., 2005*). *eat-2*; *egl-3* mutants grow slowly compared to *eat-2* due to their slow pumping rate, showing that we could use growth rate as an indicator for the pumping defect. *eat-2*; *eri-1*; *lin-15b* mutants grew slowly after *egl-3* RNAi, whereas *rrf-3*; *eat-2* did not. This suggested that *eat-2*; *eri-1*; *lin-15b* mutants are more sensitive to RNAi in the cells that produce neuropeptides relevant to MC-independent pumping. Thus we used *eat-2*; *eri-1*; *lin-15b* for the RNAi screen.

From the RNAi screen of 115 neuropeptide genes we found several that changed the growth rate of *eat-2*; *eri-1*; *lin-15b* mutants. RNAi of five NLP (Neuropeptide-Like Proteins) family genes (*nlp-2*, *nlp-12*, *nlp-14*, *nlp-24*, and *nlp-34*) and 2 INS (INSulin related) family genes (*ins-34* and *ins-39*) decreased growth rates compared to control (*Supplementary file 1*). Among these seven genes RNAi of *nlp-24* delayed growth most. To validate the RNAi results, we constructed *eat-2*; *nlp-24* double mutants and measured their pumping rate. As expected from RNAi results, the *eat-2*; *nlp-24* mutants

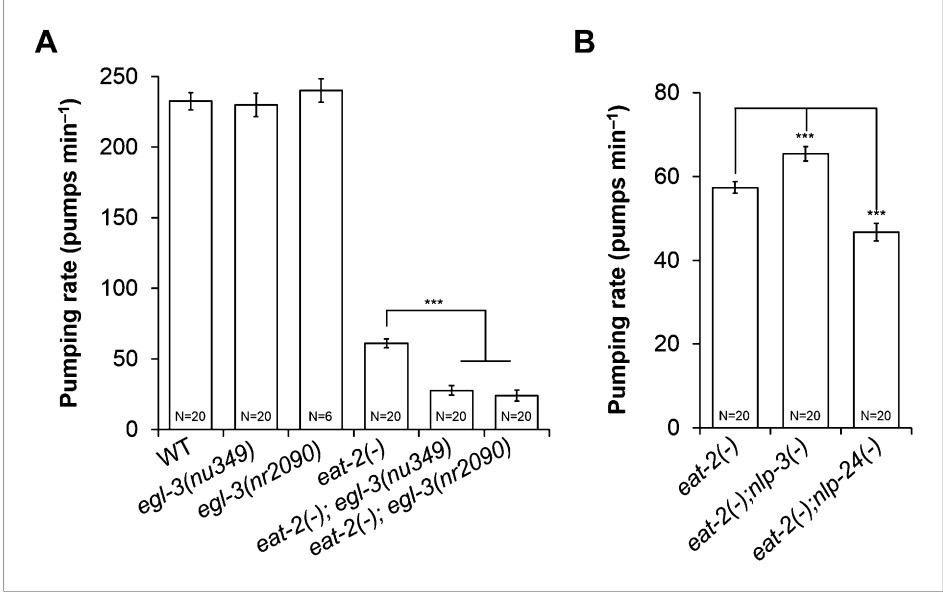

**Figure 1**. MC neurons and neuropeptides redundantly control pumping rate. (**A**) Pumping rate in *eat-2*, *egl-3* and *eat-2*; *egl-3* mutants. *egl-3* reduced pumping in *eat-2* mutants. ***p < 0.001. (ANOVA + Tukey tests). (**B**) *nlp-3* suppressed pumping and *nlp-24* stimulated pumping in *eat-2* mutants. ***Different from *eat-2*, p < 0.001 (ANOVA + Dunnett tests). In this and all figures, errors bars represent standard error.

had decreased pumping rates compared to *eat-2* (*Figure 1B*). Interestingly, RNAi of *nlp-3* and *nlp-13* improved the growth rate of *eat-2* mutants, suggesting that peptides encoded by these genes might antagonize MC-independent pumping. Indeed, the *eat-2*; *nlp-3* mutants have increased pumping rates compared to *eat-2*. These results demonstrate that *nlp-24* increases pumping and *nlp-3* suppresses pumping in the absence of MC neuron activity. Because *eat-2*; *nlp-24* mutants do not pump as slowly as *eat-2*; *egl-3* mutants (*Figure 1A,B*), we suggest that there may be other neuropeptides that act together with *nlp-24* to stimulate pumping in the absence of MC neuron activity.

Based on these results we hypothesize that *nlp-24* and *nlp-3* regulate pumping rate in the absence of food when MC is inactive.

## NLP-24 regulates pumping under starvation conditions

In mammals, one major mode of neuropeptide regulation is control of mRNA levels (*Brady et al., 1990*; *Challis et al., 2003*). Thus we measured the mRNA levels of *nlp-3* and *nlp-24* in well-fed and starved worms. *nlp-3* mRNA levels were unaffected, but the *nlp-24* increased after a 3 hr starvation, suggesting it plays a role in regulation of pumping during starvation (*Figure 2A*). Wild type worms initially pump slowly when taken off food, but increase pumping rate gradually during several hours of starvation (*Avery and Horvitz, 1990*). In confirmation, *nlp-24* mutants showed a reduced pumping rate compared to wild type after 1 hr of starvation, and this defect was rescued by a transgene that encoded the wild-type copy of *nlp-24* gene under control of its own promoter. Moreover, *nlp-24* overexpression further increased the pumping rate in starved worms compared to non-transgenic worms (*Figure 2B*). These results indicate that *nlp-24* stimulates pumping during starvation. *egl-3* mutants, which lack most active neuropeptides, also had decreased pumping during starvation (*Figure 2—figure supplement 1*), showing that neuropeptides contribute to pumping during starvation.

NLP-24 is predicted to be processed into peptides contain YGGY sequences either internally or at the N-terminus (*Nathoo et al., 2001*; *Janssen et al., 2008a*) (*Figure 2C*). Comparing the sequence of NLP-24 peptides with that of human neuropeptides, we noted that the YGGY sequences are similar to the conserved YGGF sequence motif of endogenous opioids in humans (*Snyder and Pasternak, 2003*). Thus we hypothesize that *nlp-24* potentially encodes endogenous opioids in *C. elegans*. Although no *C. elegans* opioid signaling system has been reported, a study identified NPR-17, a G protein coupled receptor, as most closely related to a predicted *Brugia malayi* ORL1 opioid receptor-like protein with 68% identity (*Harris et al., 2010*). They found that *nlp-3*, a neuropeptide gene, and *npr-17* regulated avoidance behavior, suggesting NLP-3 and NPR-17 together could mediate a signal to control pain in *C. elegans* (*Harris et al., 2010*). In BLAST searches, we verified their finding further; NPR-17 is similar to human nociceptin receptor (28% identity), MOR (23% identity), DOR (23% identity) and to the KOR (24% identity) (*Figure 2—figure supplement 2*). Together these results suggested that NLP-24 peptides and NPR-17 receptor constitute an endogenous opioid system in *C. elegans*.

## Opioids modulate pumping through NPR-17, a homolog of mammalian opioid receptors

Because NLP-24 stimulated *C. elegans* pumping under starvation conditions and because previous work and sequence homology suggested NPR-17 could be an opioid receptor, we tested whether opiates affect pumping, and whether NPR-17 mediates these effects. First, we treated starved wild type and *npr-17* mutants with opioid agonist morphine and blocker naloxone and measured the pumping rate. As predicted, morphine stimulated pumping whereas naloxone suppressed pumping in wild type. Drugs get into intact worms inefficiently, both because the wild-type cuticle is poorly permeable (*Partridge et al., 2008*) and because foreign chemicals are actively pumped out (*Broeks et al., 1996*; *Ardelli and Prichard, 2013*). For instance, although 1 µM serotonin maximally stimulates pumping in dissected pharynxes (*Niacaris and Avery, 2003*), roughly 10 mM is necessary to produce similar effects in intact animals (*Horvitz et al., 1982*; *Hobson et al., 2006*; *Raizen et al., 2012*). Thus we were not surprised that high concentrations of morphine (0.5 mM) and naloxone (10 mM) were necessary to produce these effects. Specificity was demonstrated by the observation that these effects were completely abolished by *npr-17* mutations, showing that NPR-17 mediated the effects, suggesting it could be an opioid receptor in *C. elegans* (*Figure 3A,B*). We tested two different

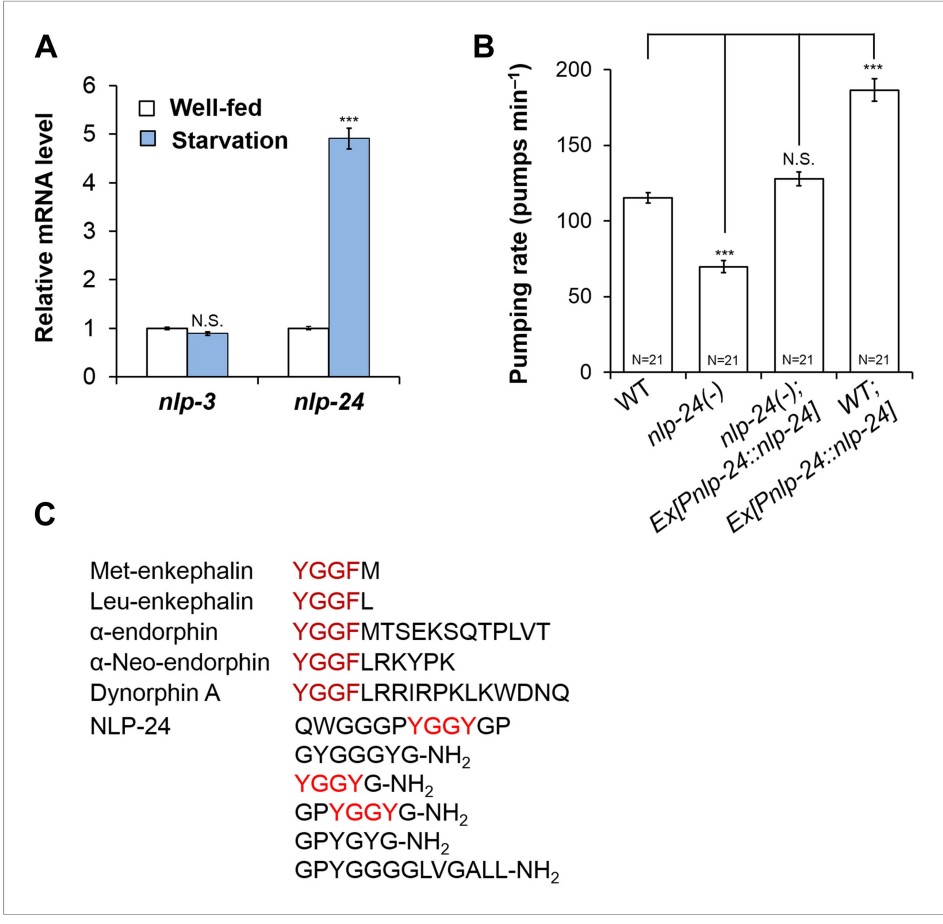

**Figure 2**. *nlp-24* stimulates pumping in starved worms. (**A**) *nlp-3* and *nlp-24* mRNA levels in well-fed and starved worms. qRT-PCR was performed to assess mRNA expression. Normalized with *ama-1*. ***Different from well-fed, $p < 0.001$ (two-way ANOVA on $C_t$ with Bonferroni correction). (**B**) Pumping rate after 1 hr starvation. *nlp-24* stimulated pumping during starvation. ***$p < 0.001$, N.S. not significant (ANOVA + Tukey tests). (**C**) NLP-24 and human endogenous opioid peptides.

The following figure supplements are available for figure 2:

**Figure supplement 1**. Neuropeptides stimulate pumping in starvation.

**Figure supplement 2**. Phylogenetic analysis of opioid receptors.

**Figure supplement 3**. *nlp-3* pumping rate after 1 hr starvation.

**Figure supplement 4**. Well-fed *nlp-24* mutant pumping rate.

alleles (*tm3210* and *tm3225*) of *npr-17* and both alleles gave the same results (*Figure 3B* and *Figure 3—figure supplement 1*). Taken together these results indicate that opioids stimulate *C. elegans* pumping and that NPR-17 might be a worm opioid receptor.

To further test whether *npr-17* mediates *nlp-24* action, we overexpressed *nlp-24* in *npr-17* mutants. *nlp-24* overexpression in the wild type background caused faster pumping under starvation conditions as shown above. However, *nlp-24* overexpressed in the *npr-17* background did not increase the pumping rate (*Figure 4A*). *nlp-24*; *npr-17* double mutants also pumped at the same rate as *npr-17* mutants (*Figure 4B*). These results indicate that NPR-17 is downstream of NLP-24 and are consistent with the hypothesis that NPR-17 is a receptor for NLP-24-derived peptides to regulate pumping during starvation.

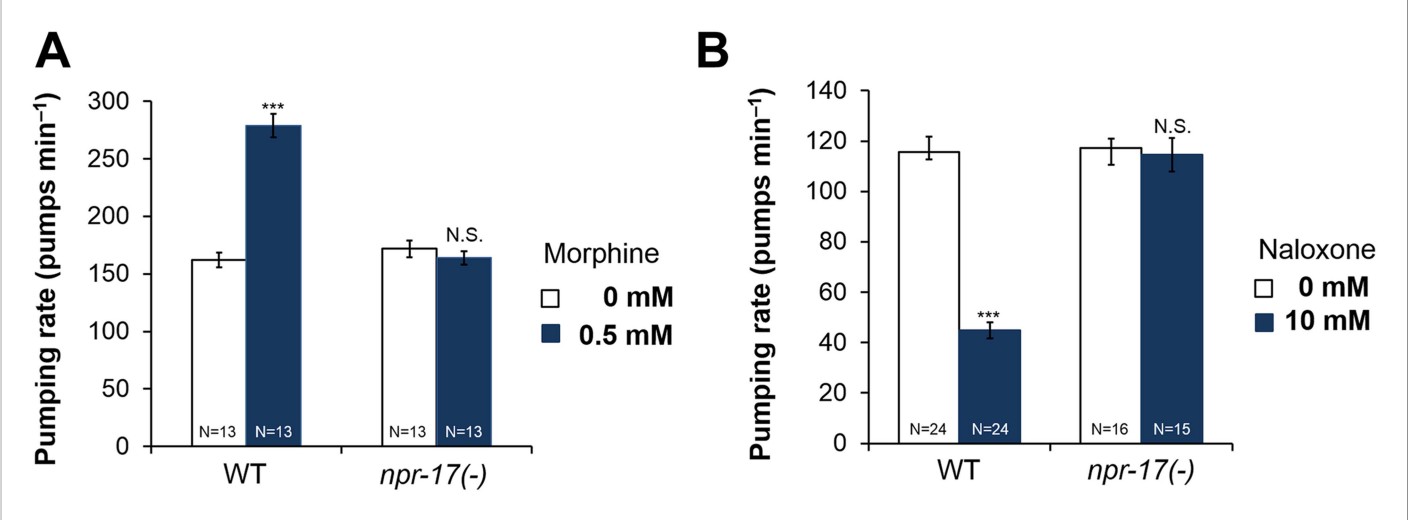

**Figure 3**. Opioids control pumping through *npr-17*. (**A**) Effect of morphine on pumping. (**B**) Effect of naloxone on pumping rate. Adult hermaphrodites were starved for 1 hr and tested. Morphine stimulates pumping and naloxone inhibits it in wild-type worms; neither drug affects *npr-17(tm3210)* mutants. ***Different from 0 mM, p < 0.001; N.S. not significantly different from 0 mM (two-way ANOVA, concentration effect). The effects of morphine and naloxone on wild-type are significantly different from their effects on *npr-17*, p < 0.001 (two-way ANOVA, genotype × concentration interaction).

The following figure supplements are available for figure 3:

**Figure supplement 1**. Effect of naloxone on pumping rate in *npr-17(tm3225)* mutant worms.

**Figure supplement 2**. Naloxone reduces the pumping rate of *eat-2* mutants.

Next we examined whether NLP-24 peptides indeed function as opioids and NPR-17 as an opioid receptor using heterologous expression in HEK293 cells. We coexpressed NPR-17 in HEK293 cells with the promiscuous G protein Gα15 and administered synthetic NLP-24 peptides. In this heterologous system, only one of these peptides, GPYGYGamide, activated NPR-17 (*Figure 5A*), suggested that NLP-24 encodes at least one NPR-17 ligand. We also tested agonists specific for mammalian opioid receptor subtypes: loperamide for the MORs (*DeHaven-Hudkins et al., 1999*), SB205607 for the DORs (*Fujii et al., 2004*), and U69593 for the KORs (*Towett et al., 2006*). NPR-17 was also activated by loperamide and U69593 (*Figure 5A*). GPYGYGamide, loperamide, and U69593 activated NPR-17 in a dose-dependent manner, and these responses were abrogated by naloxone (*Figure 5B*). These results indicate that NPR-17 is an opioid receptor and NLP-24 peptide GPYGYGamide is a *C. elegans* opioid.

We then tested whether NLP-24 peptides can activate human opioid receptors. We individually coexpressed each human opioid receptor in HEK293 cell with Gα15. We tested an agonist specific for each receptor subtype in addition to NLP-24 peptides. Mu-type opioid receptor (MOR-1) and kappa-type opioid receptor (KOR-1) responded only to YGGYGamide peptide, while delta-type opioid receptor (DOR-1) did not respond to any NLP-24 peptide (*Figure 6*). Thus *C. elegans* has a peptide that can activate human opioid receptors, predicted to be produced from the same precursor that produces at least one ligand for the *C. elegans* opioid receptor NPR-17, which can also be activated by vertebrate receptor-specific opioid agonists. These results suggested that *C. elegans* and vertebrates share a conserved opioid system.

## ASI neurons are required for opioid signaling and NPR-17 in ASI neurons is sufficient for opioid signaling

*npr-17* was shown to be expressed in ASI neurons (*Harris et al., 2010*), a pair of head sensory neurons that regulate multiple food-related effects such as quiescence, dauer formation, and caloric-restriction-induced life span increase (*Inoue and Thomas, 2000*; *Bishop and Guarente, 2007*; *Gallagher et al., 2013b*). Because the pumping rate regulated by *nlp-24* is also food-related and the receptor NPR-17 is known to be expressed in ASI, we tested whether ASI neurons are important for opioid signaling. When we treated worms whose ASI neurons were genetically ablated with naloxone,

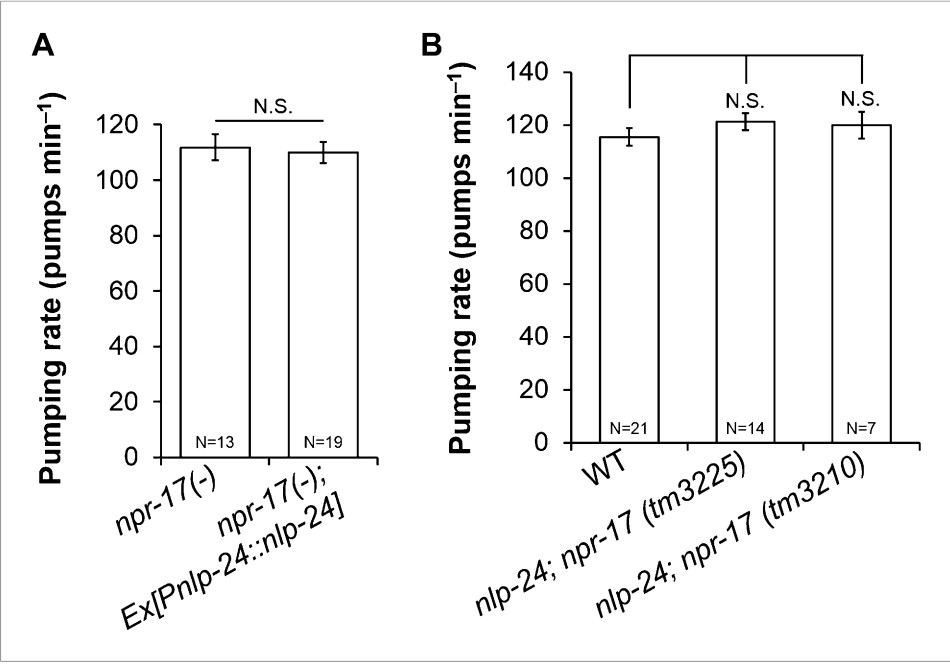

**Figure 4**. *nlp-24* regulates pumping through *npr-17*. (**A**) Overexpressed *nlp-24* in *npr-17* mutants did not stimulate pumping. (**B**) In the *npr-17* background, *nlp-24* mutations have no effect on pumping rate in starved worms. (N.S. not significant, Student's *t*-test).

there was no naloxone-induced reduction in pumping (*Figure 7A*). Thus the opioid blocker controlled pumping through ASI neurons. Next, to examine whether ASI neurons are the site of action of NPR-17, we expressed it in ASI using the ASI specific promoter, *gpa-4*. Expression of NPR-17 only in the ASI neurons completely restored the pumping reduction induced by naloxone treatment (*Figure 7B* and *Figure 7—figure supplement 1*). *npr-17* is also expressed in intestine. We therefore also tested the effect of naloxone on worms engineered to express NPR-17 only in the intestine (using the *ges-1* promoter). These worms were not rescued (*Figure 7B* and *Figure 7—figure supplement 1*). Thus NPR-17 in ASI mediated opioid signaling to regulate pumping during starvation.

While the insensitivity of *npr-17* mutants to naloxone is consistent with NPR-17 being the naloxone receptor, the sign of the effect was initially surprising. In the absence of drug, *npr-17* mutants pumped at the same rate as wild type, while they pumped faster than wild-type worms in the presence of naloxone. This would suggest that naloxone is an agonist rather than a blocker, but this is inconsistent with the known effect of naloxone of opioid receptors, as well as the measured effect on heterologously expressed NPR-17 (*Figure 5B*). An alternate hypothesis is that *npr-17* worms, having lacked the receptor all their lives, had adapted to its absence. One of these adaptations was an increase in the basal pumping rate. In wild-type worms, in contrast, NPR-17 had always been around supporting basal pumping. When we suddenly removed this support by adding naloxone, the pumping rate went down. This hypothesis predicts that the chronic effects of naloxone on wild type should resemble those of the mutation. We thus grew wild-type worms to adulthood on naloxone, then measured the pumping rate after starvation either with or without naloxone. As predicted, wild-type worms grown and tested in the presence of naloxone pumped at the same rate as worms grown and tested without (*Figure 7C*). Furthermore, after chronic naloxone treatment worms tested in its absence showed a withdrawal effect: they pumped slightly faster. These results are consistent with the hypothesis that, in vivo as in vitro, naloxone acts as an NPR-17 blocker.

*npr-17* is expressed in ASI neurons and *nlp-24* is also expressed in ASI neurons (*Figure 7—figure supplement 2C*) (*Nathoo et al., 2001*). This suggests that NLP-24 peptides may be secreted by and act on ASI neurons in an autocrine manner. We also suspect that NLP-24 is secreted by other cells and acts hormonally on ASI neurons, because, NLP-24::GFP expression is increased in intestine after starvation (*Figure 7—figure supplement 3*). An NLP-24::mCherry fusion protein accumulates in coelomocytes (*Figure 7—figure supplement 2B*), specialized cells for endocytosis and degradation

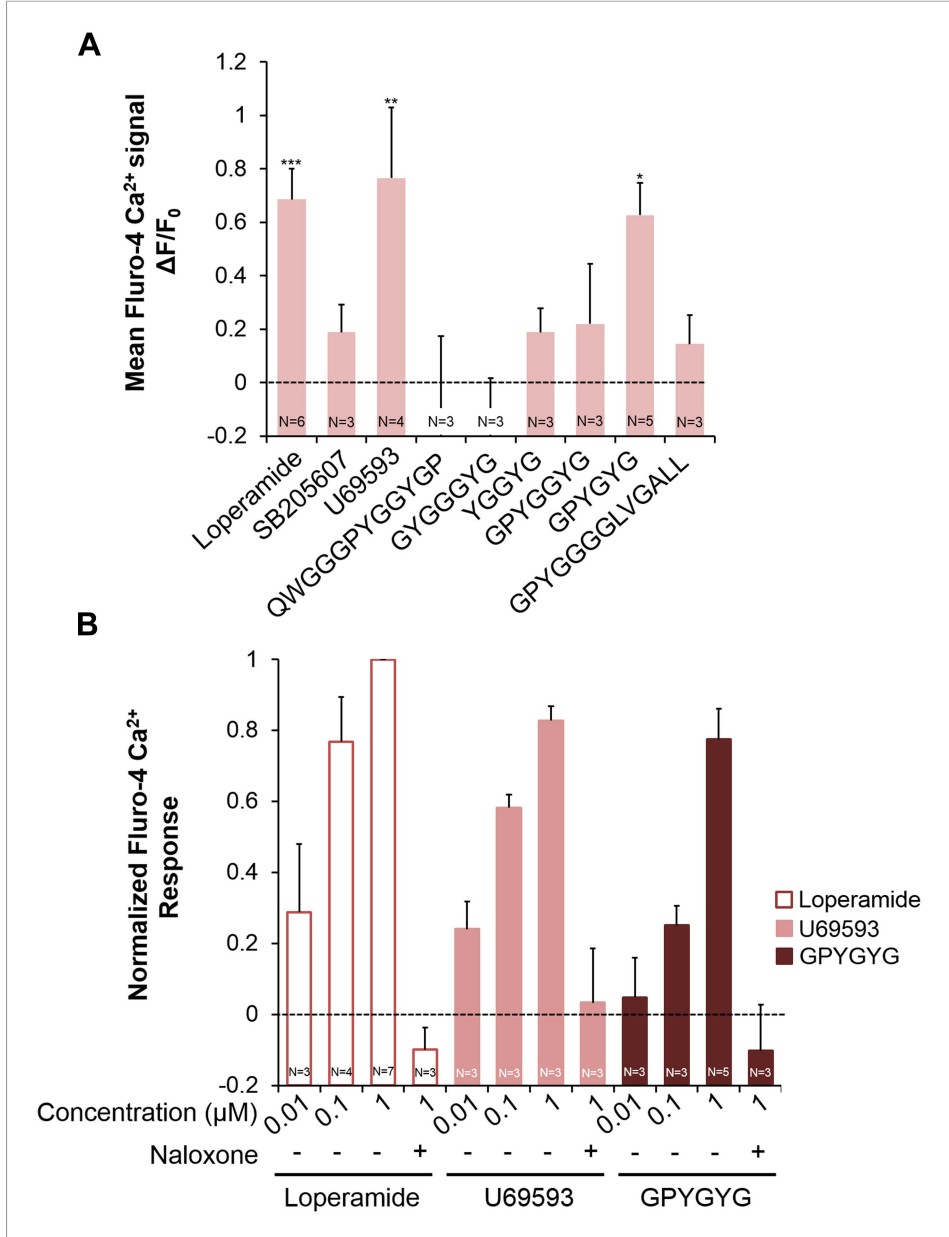

**Figure 5**. Activities of NLP-24 peptides and opioid agonists on NPR-17. Opioid agonists and NLP-24 peptide-mediated changes in intracellular calcium were measured with the calcium detector Fluo-4. (**A**) Calcium responses in human embryonic kidney 293 (HEK-293) cells transfected with NPR-17 and Gα15. All compounds were tested at 1 μM. Loperamide (μ agonist), SB205607 (δ agonist), U69593 (κ agonist) and GPYGYGamide activated NPR-17. Different from 0, *p < 0.05, **p < 0.01, ***p < 0.001, two-way ANOVA + sequential Bonferroni correction. (**B**) Dose-response curves of HEK-293 cells transfected with NPR-17 and Gα15. Loperamide, U69593 and NLP-24 peptide GPYGYGamide induced NPR-17 activation in a dose dependent manner, and these responses were suppressed by opioid blocker naloxone (10 μM). Responses were normalized to 1 μM loperamide. The loperamide effect is significant at p < 0.001, and the effects of U69593 and GPYGYGamide at p < 0.01, two-way ANOVA + sequential Bonferroni correction. Pooling the results from **A** and **B**, the effect of GPYGYGamide is significant at p < 0.001 (Stouffer's weighted Z-score test).

of secreted proteins (*Fares and Grant, 2002*), which have been found to accumulate other secreted peptide signals (*Sieburth et al., 2007*; *Lee and Ashrafi, 2008*). Our result is thus consistent with the idea that NLP-24 is a precursor for secreted peptides, supporting the idea that NLP-24 is secreted and sensed by NPR-17 in ASI neurons.

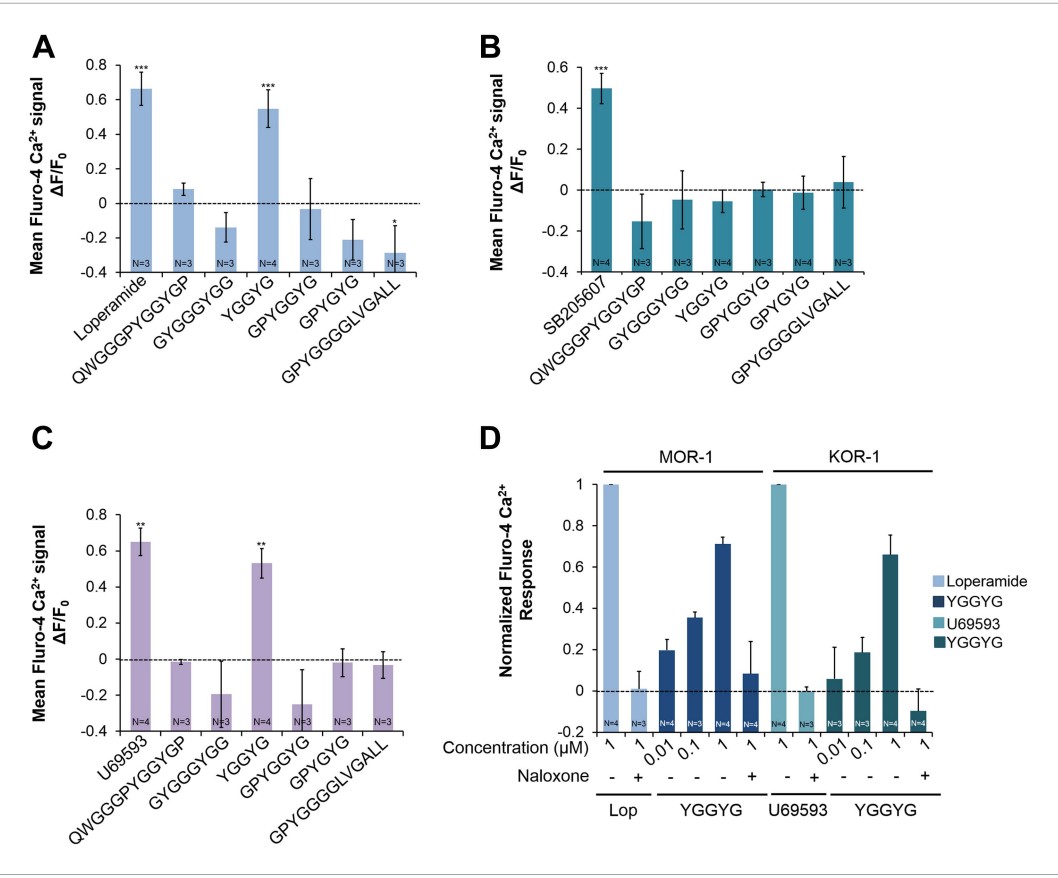

**Figure 6**. Activities of NLP-24 peptides and agonists on MOR-1, DOR-1 and KOR-1. Opioid agonist and NLP-24 peptide-mediated changes in intracellular calcium were measured by the calcium detector Fluo-4. Agonists and peptides were tested at 1 μM. (**A**) Calcium responses in HEK-293 cell transfected with MOR-1 and Gα15. (**B**) Calcium responses in HEK-293 cell transfected with DOR-1 and Gα15. (**C**) Calcium responses in HEK-293 cell transfected with KOR-1 and Gα15. ***p < 0.001, **p < 0.01, *p < 0.05. (Two-way ANOVA with Bonferroni correction.) (**D**) Dose-response curves of HEK-293 cells transfected with MOR-1 or KOR-1 and Gα15. NLP-24 peptide YGGYGamide activated MOR-1and KOR-1, and these responses were suppressed by the opioid blocker naloxone (10 μM). MOR-1 responses were normalized to 1 μM loperamide and KOR-1 to 1 μM U69593. The effects of YGGYGamide on MOR-1 and KOR-1 are significant at p < 0.001 (two-way ANOVA).

Together our results demonstrate that *C. elegans* has an endogenous opioid system, which modulates pumping during starvation. But what is the purpose of pumping in the absence of food? Following a suggestion by David Raizen (personal communication), we suggest that this pumping helps worms discover food in the environment, even when their external senses are unable to reliably detect it. According to this hypothesis, opioids help worms survive starvation by increasing the probability of catching food. In essence, food-independent pumping is (or facilitates) exploration. This is consistent with the finding that in many animals, including *C. elegans*, starvation increases exploration (*Dallman et al., 1999*; *Torres et al., 2002*).

This hypothesis suggested that, in addition to increasing pumping to enhance the chance of catching food, opioids might increase locomotion to enhance the chance of finding food. *C. elegans* locomotion is tightly related to nutritional status (*Shtonda and Avery, 2006*; *Ben Arous et al., 2009*). There are three potentially distinct locomotive states: roaming, dwelling, and quiescence (*Fujiwara et al., 2002*; *You et al., 2008*). Roaming is the food-seeking behavior; worms roam when the food is scarce. We therefore tested the effect of *nlp-24* mutation on locomotion in starved worms. Wild type spent 93% of the time roaming, but in *nlp-24* mutants this was reduced to 69% (*Figure 8*). An *nlp-24* transgene rescued the defect. Based on these results, we suggest that by increasing pumping and increasing exploration opioids help worms survive starvation.

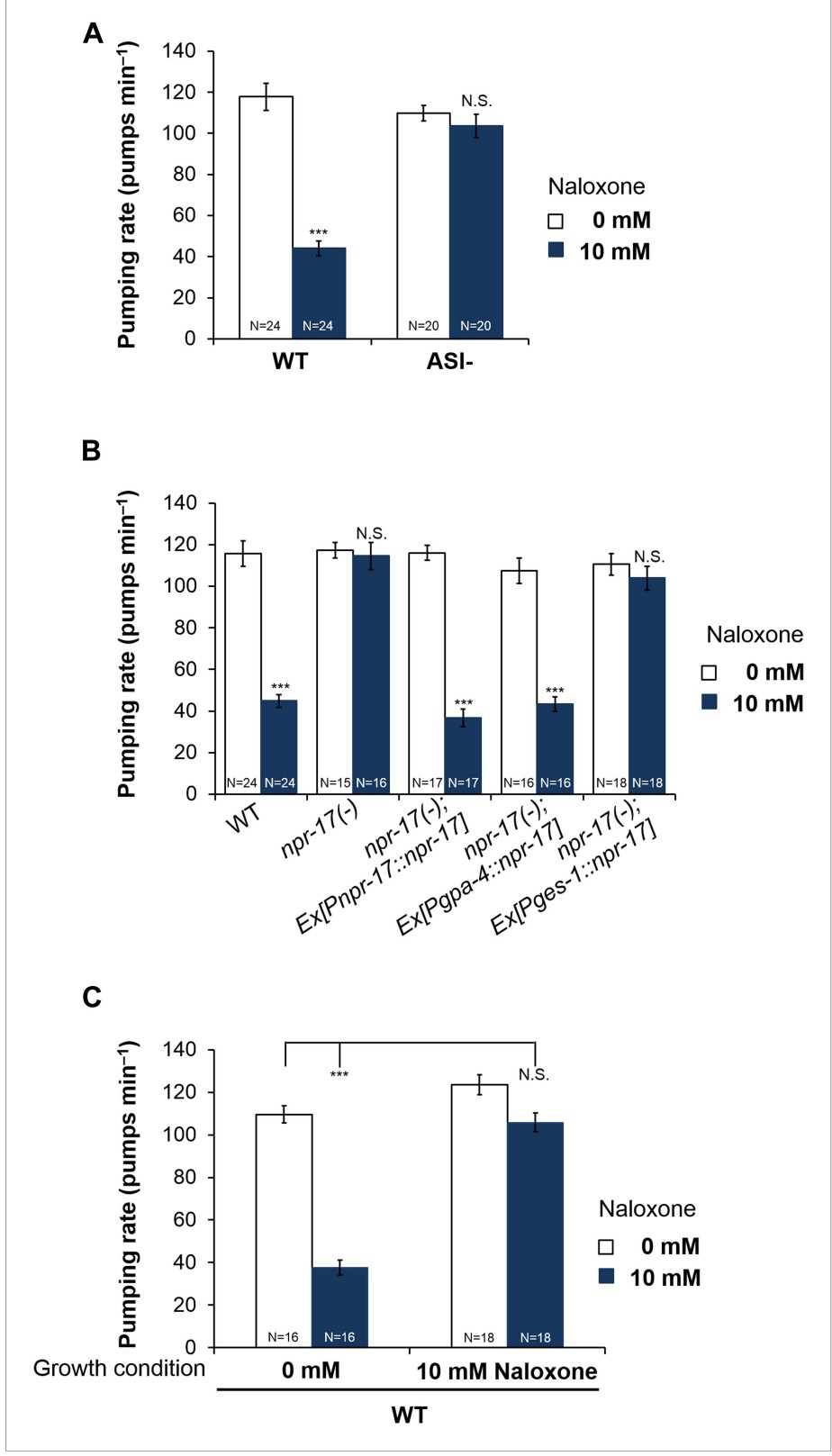

**Figure 7**. ASI neurons are required for opioid-mediated feeding control. (**A**) Effect of naloxone on pumping. Worms genetically engineered to lack ASI neurons were not affected. ASI neurons were genetically ablated by the recCaspase method (**Chelur and Chalfie, 2007**; **Beverly et al., 2011**) (ASI−) ***Different from 0 mM, p < 0.001; N.S. *Figure 7. continued on next page*

*Figure 7. Continued*

not significantly different from 0 mM (two-way ANOVA, concentration effect). The effect of naloxone on wild-type is significantly different from its effect on *npr-17*, p < 0.001 (two-way ANOVA, genotype × concentration interaction). (**B**) *npr-17* expression in ASI neurons is required for opioid-mediated feeding control. *npr-17* mutants were not affected by naloxone, but this phenotype was rescued with by expression of *npr-17* under control of the ASI-specific promoter *gpa-4*. ***Different from 0 mM, p < 0.001, N.S., not significantly different from 0 mM (two-way ANOVA, concentration effect). The interaction between genotype and concentration is significant at p < 0.001 for *Pges-1::npr-17* rescued vs WT, but not *Pnpr-17::npr-17* or *Pgpa-4::npr-17* rescued. The interaction between genotype and concentration is significant at p < 0.001 for *Pnpr-17::npr-17* and *Pgpa-4::npr-17* rescued vs *npr-17*, but not *Pges-1::npr-17* rescued. (**C**) Chronic effect of naloxone on pumping. Worms grown to adulthood with or without 10 mM naloxone plate were then tested with or without 10 mM naloxone. ***p < 0.001; N.S. not significantly different. The stimulatory effect of taking naloxone away from worms grown in the presence of naloxone is significant at p < 0.05. (ANOVA + Tukey tests).

The following figure supplements are available for figure 7:

**Figure supplement 1**. *npr-17* in ASI neurons are required for opioid-mediated feeding control.

**Figure supplement 2**. *nlp-24* expression.

**Figure supplement 3**. Starvation induces *nlp-24* expression in intestine.

## Discussion

We hope as a result of these studies to establish *C. elegans* as an invertebrate genetic model for the study of opioid signaling. Our results confirm previous indications that opioids have a long evolutionary history. Several researchers have found evidence that the opioid system might exist in invertebrates: peptides encoding opioids or opioid receptors were detected, and drugs such as naloxone that act on opioid receptors induced biological responses (*Harrison et al., 1994*). We confirm these results in *C. elegans* and add a crucial control: a mutant lacking the receptor fails to respond. Furthermore, like vertebrate opioids, worm opioids have a role in feeding behavior. *nlp-24* encodes endogenous opioids and *npr-17* an opioid receptor, and together they act to regulate feeding behavior. Opioids also participate in the immune system and pain mechanisms in vertebrates (*McCarthy et al., 2001*). *nlp-24* and *npr-17* may have related functions in *C. elegans*. *nlp-24* expression is down regulated by a pathogenic bacterium, *Pseudomonas aeruginosa*, and a fungus, *Drechmeria coniospora* (*Troemel et al., 2006*; *Pujol et al., 2008*). *npr-17* mediates ASH-mediated aversive behavior (*Harris et al., 2010*), a *C. elegans* model for pain.

*Harris et al. (2010)* found that *nlp-3* affects ASH-mediated aversion upstream of NPR-17, and suggested therefore that NPR-17 might be an NLP-3 peptide receptor. We too found that *nlp-3* mutation affects MC-independent pumping, although it had no effect on starved wild-type worms (*Figure 2—figure supplement 3*). While we cannot exclude the possibility that NLP-3 peptides are NPR-17 ligands, Harris et al.'s results are equally well explained by a model in which NLP-3 peptides affect NPR-17 indirectly by affecting the release of NLP-24 peptides.

We showed that opioids stimulate pumping during starvation. We started with the hypothesis that neuropeptides regulate pumping in the absence of MC activity. Thus, we performed experiments in starved worms. Opioid agonists and antagonists modulated pumping in starved worms, and the opioid mutant *nlp-24* had decreased pumping in starvation. In the presence of food, in contrast, both *nlp-24* mutants (*Figure 2—figure supplement 4*) and *egl-3* mutants (*Figure 1A*) showed the same pumping rate as wild type. Nutritional state affected opioid activity: *nlp-24* expression was increased by starvation (*Figure 2A*). Likewise, food deprivation and feeding changes endogenous opioid activity in mammals. In rats, food deprivation induces complex changes in the level of brain endogenous opioids, especially in the hypothalamus (*Vaswani and Tejwani, 1986*). In addition, opioid signaling may be involved in altering the hedonic taste/palatability of food in mammals (*Bodnar, 2004*).

Why does *C. elegans* need opioid signaling to control feeding during starvation? In a well-fed worm, pumping depends on MC motor neurons, which are active only in the presence of food.

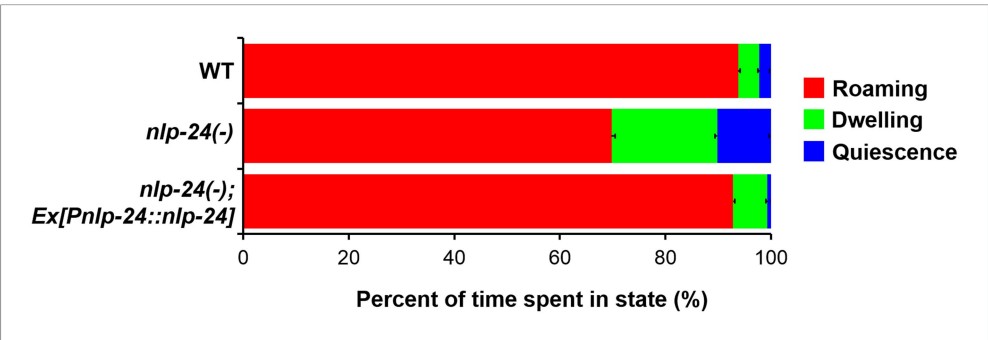

**Figure 8**. *nlp-24* stimulates roaming behavior in starved worms. Behavioral states of adult worms on plates without food. A single worm was starved for 30 min, then recorded for 1 hr. Locomotion was analyzed as described previously (Gallagher et al., 2013). Starvation increased roaming behavior in wild type. *nlp-24* mutants had reduced roaming behavior compared to wild type, and an *nlp-24* transgene rescued this phenotype. N = 22 worms for wild type, 27 for *nlp-24* mutants, and 19 for *nlp-24* rescued worms. All *nlp-24* state probabilities are significantly different from the corresponding probabilities for wild type and the rescued strain, p < 0.001, with the exception of the Q state for *nlp-24* vs wild type, for which p < 0.01. There are no significant differences between wild type and *nlp-24* rescued. (Mann–Whitney U-test with sequential Bonferroni correction).

But opioids modulated pumping independent of MC neurons: naloxone decreased pumping in *eat-2* mutants, which lack MC neuromuscular transmission (*McKay et al., 2004*) (*Figure 3—figure supplement 2*). We speculate that, like a baby whose first response to any new object is to put it in its mouth, the worm cannot be sure if there is food around without taking some in. A well-fed worm avoids this dangerous test, but a starving worm continually pumps at a low rate even in the absence of food in order to detect any that might appear. Consistent with this, we found that opioids motivate worms to seek food by stimulating locomotion as well as pumping.

Thus opioids make worms willing to search for food despite the high risk. Animals live in dynamic environments where food availability is variable. For survival in this environment, food-seeking behavior is important. Opioids help worms find food through risky pumping and exploration behaviors. Thus endogenous opioids may help worms survive in dynamic environments.

Opioids stimulated pumping through ASI neurons (*Figure 7*). ASI neurons are activated by nutrition and release DAF-7 TGF-β when the environment is favorable. *daf-7* expression in ASI neurons regulates fat accumulation and food intake (*Greer et al., 2008*). ASI neurons also stimulate roaming behavior (*Flavell et al., 2013*). These results suggest that ASI neurons are important for modulating energy balance and feeding behavior. In mammals opioid receptors are expressed in the hypothalamus, which plays a central role in the control of food intake and the regulation of energy. Injection of opioid agonists or antagonists into the hypothalamus modulates food intake (*Jenck et al., 1987*; *Li et al., 2006*; *Naleid et al., 2007*). We suggest that opioids modulate *C. elegans* feeding through ASI neurons, similar to the way vertebrate opioids modulate feeding through the hypothalamus.

Which pharyngeal neurons are involved in opioid signaling? Our lab did several suppressor screens with slow pumping. From these screens, the most commonly mutated gene was *slo-1*. *slo-1* mutations enhance neurotransmitter release (*Wang et al., 2001*). This enhancement is particularly striking for M4 and M5, the motor neurons that synapse on TB (terminal bulb) muscle (*Albertson and Thomson, 1976*; *Chiang et al., 2006*). We also screened for suppressors of the slow pumping and growth phenotypes of *eat-2*; *egl-3* mutants. 7/15 suppressors were new *slo-1* mutations. These results are consistent with the possibility that M4 might be involved in the opioid signaling response. We do not know yet how ASI neurons communicate with pharyngeal neurons. The pharyngeal and extrapharyngeal nervous systems are connected only by a bilateral pair of gap junctions between the extrapharyngeal RIP neurons and the pharyngeal I1 neurons (*Albertson and Thomson, 1976*), and I1s are not necessary for MC-independent pumping (*Avery and Horvitz, 1989*). This, together with the fact that ASI secretes several different peptides, suggests that communication is likely to be humoral.

Heterologously expressed NPR-17 was activated by specific MOR-1 and KOR-1 agonists. These results suggested that NPR-17 might be more similar to μ and κ type receptors than δ. We also tested

pumping with these agonists. Among the three, loperamide induced nose muscle contraction similar to that caused by fluoxetine (*Choy and Thomas, 1999*), so that we could not measure the pumping rate. U69593 increased pumping in wild-type worms but not *npr-17* mutants (*Figure 9*). These results suggest that NPR-17 might be similar to the KOR but do not exclude similarity to μ as well. Vertebrate opioid receptors are about 60% identical to each other (*Chen et al., 1993*). In BLAST results, NPR-17 showed 23% identity with MOR-1 and DOR-1 and 24% identity with KOR-1. The three vertebrate receptors almost certainly diverged from each other well after any common ancestor of nematodes and vertebrates, so NPR-17 is probably an ortholog of all three. Clearly, NPR-17 is not a close homolog of any particular vertebrate opioid receptor. Rather, we suggest that NPR-17 and the vertebrate receptors collectively derive from a common ancestor in which the outlines of opioid system function had already appeared. Further, NPR-17 appears to be functionally closer to vertebrate opioid receptors than any other molecularly characterized invertebrate receptor is.

We originally thought that NLP-24 might be related to opioids because it encodes peptides containing YGGY, similar to the YGGF sequence common to vertebrate opioids. In fact, of the six predicted NLP-24 peptides, YGGYGamide alone activated MOR-1 and KOR-1 expressed in human embryonic kidney 293 (HEK-293) cells. But, to our surprise, YGGYGamide did not detectably activate NPR-17 in the same system. In fact, the only peptide that worked was GPYGYGamide. By itself, this result might cast doubt on the idea that NLP-24 and NPR-17 are components of the *C. elegans* opioid system. The extreme version of this doubt would argue that NLP-24, like POMC, is a precursor for multiple peptides that act on distinct unrelated receptors, and that it is essentially coincidence that the NPR-17 ligand GPYGYGamide shares a precursor with the opioid YGGYGamide. But this idea fails to explain multiple lines of data suggesting NPR-17 is a bona fide opioid receptor, namely: (1) Heterologously expressed NPR-17 responds to opioid receptor agonists and the blocker naloxone. (2) Naloxone blocks the response of NPR-17 to GPYGYGamide. (3) *npr-17* is necessary for the behavioral response of worms to opioid agonists and naloxone. (4) NPR-17 has sequence similarity to opioid receptors. A more parsimonious explanation of the discrepancy is that in the HEK-293 expression system, where we asked a single *npr-17* product to work entirely through vertebrate proteins, we failed to completely reconstitute its in vivo activity. Of course, these ideas are not mutually exclusive. It may be that in the worm some form of NPR-17 does respond to YGGY-containing NLP-24 products, and that NLP-24 produces products that act on other receptors as well.

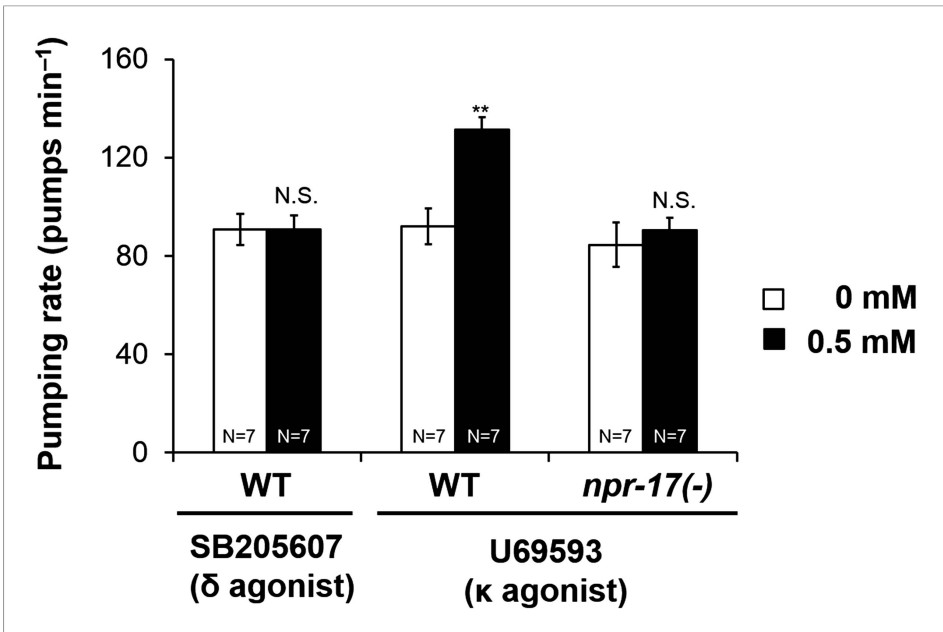

**Figure 9**. Opioid agonist effects in *C. elegans*. The δ agonist SB205607 did not affect pumping. κ agonist U69594 stimulated pumping in starved wild-type worms but not *npr-17* mutants. **p < 0.01. N.S. not significant. (Two-way ANOVA).

Our study confirms with molecular genetic data the previous pharmacological evidence that an invertebrate homolog of the vertebrate opioid system exists and has related functions. In the future, we hope this will enable the discovery of new molecular pathways through which opioids act and perhaps help to find ways to mitigate the side-effects that limit the usefulness of opiate drugs.

## Materials and methods

### Cultivation methods and strains

Worms were cultured routinely on NGMSR plates (*Avery, 1993*). All worms were maintained at 20°C on *Escherichia coli* strain HB101 unless indicated otherwise. The wild-type strain was *C. elegans* variant Bristol, strain N2. The following mutant strains were used in this study: DA465 *eat-2(ad465) II*, GR1328 *egl-3(nr2090) V*, KP1873 *egl-3(nu349) V*, DA2434 *eat-2(ad465) II*; *egl-3(nu349) V*, DA2435 *eat-2(ad465) II*; *egl-3(nr2090) V*, DA2467 *eat-2(ad465) II*; *nlp-3(tm3023) X*, DA2456 *nlp-24(tm2105) V*, DA2466 *eat-2(ad465) II*; *nlp-24(tm2105) V*, DA2457 *npr-17(tm3210) III*, DA2458 *npr-17(tm3225) III*, DA2561 *adEx2561[npr-17::GFP Punc-122::RFP]*, DA2582 *npr-17(tm3210) III*; *adEx2561[Pnpr-17::npr-17::GFP Punc-122::RFP]*, DA2583 *npr-17(tm3225) III*; *adEx2561[Pnpr-17::npr-17::GFP Punc-122::RFP]*, DA2586 *npr-17(tm3210) III*; *adEx2586[Pgpa-4::npr-17::SL2::RFP Punc-122::RFP]*, DA2587 *npr-17 (tm3210) III*; *adEx2587[Pges-1::npr-17::SL2::RFP Punc-122::RFP]*, DA2588 *npr-17(tm3225) III*; *adEx2588[Pgpa-4::npr-17::SL2::RFP Punc-122::RFP]*, DA2589 *npr-17(tm3225) III*; *adEx2589[Pges-1:: npr-17::SL2::RFP Punc-122::RFP]*, DA2596 *npr-17(tm3210) III*; *adEx2596[Pnlp-24::nlp-24::GFP Punc-122::RFP]*, DA2597 *npr-17(tm3225) III*; *adEx2597[Pnlp-24::nlp-24::GFP Punc-122::RFP]*, DA2557 *adEx2557[Pnlp-24::nlp-24::mCherry Punc-122::GFP]*, DA2590 *adEx2590[Pnlp-24::nlp-24::GFP Punc-122::RFP]*, DA2591 *adEx2591[Pnlp-24::nlp-24::GFP Punc-122::RFP]*, DA2592 *adEx2592[Pnlp-24::GFP Punc-122::RFP]*, DA2593 *nlp-24(tm2105) V*; *adEx2593[Pnlp-24::nlp-24::GFP Punc-122::RFP]*, PY7505 *oyIs84[Pgpa-4::TU813 Pgcy-27::TU814 Pgcy-27::eGFP Punc-122::DsRed]* (*Beverly et al., 2011*) PY7005 was gift from Piali Sengupta.

### Molecular biology and transgenesis

Most DNA constructs were made using overlap-extension PCR (*Hobert, 2002*). Briefly, to fuse two or more individual PCR products (PfuUltra High-Fidelity DNA Polymerase, stratagene, La Jolla, CA), we amplified with oligonucleotides that included 5′ extensions complementary to the fusion target. Primers for making these constructs are listed in *Supplementary file 2*. A transcriptional reporter for *nlp-24* was made by fusing the promoter of *nlp-24* (−2262 to +1) to the coding sequence of GFP amplified from pPD95.77 (*Fire et al., 1990*). For an *nlp-24* translational reporter (*Pnlp-24::nlp-24::GFP*), the genomic sequence of *nlp-24* (−2760 to +316) was fused in frame to GFP followed by the *unc-54* 3′ UTR from pPD95.77. For *nlp-24* rescue and overexpression, we used an *nlp-24* translational reporter. To make NLP-24 fused with mCherry, the genomic sequence of *nlp-24* (−2760 to +316) was fused in frame to mCherry followed by the *unc-54* 3′ UTR from pAV1997 (*Miedel et al., 2012*). To make NLP-24::SL2:: GFP, we used a pJG7-psm-SL2-GFP vector (a gift from Dr Cori Bargmann). The genomic sequence of *nlp-24* (−3151 to +460) was ligated into the pJG7-psm-SL2-GFP vector. To make an *npr-17* translational reporter (*Pnpr-17::npr-17::GFP*), we amplified GFP from pPD95.77 and inserted this PCR product in the genomic sequence of *npr-17* (−5110 to +5545) before the stop codon, thus fusing GFP with the *npr-17* 3′ UTR (+5548 to +6524). For site specific expression of npr-17, we used a vector from Douglas S Kim based on pSM into which was inserted SL2::RFP and the *unc–54* 3′ UTR. First, *npr-17* cDNA was amplified and ligated into the pSM-SL2::RFP vector as an XmaI/NheI fragment. Then, to express *npr-17* in the ASI neurons, 2.5 kb of *gpa-4* promoter amplified from N2 genomic DNA was cloned into this pSM *npr-17* SL2::RFP construct using BamHI and NotI. To express *npr-17* in the intestine, 3 kb of *ges-1* promoter was amplified from N2 genomic DNA and inserted into the pSM *npr-17* SL2::RFP construct using BamHI and NotI sites. To express in mammalian cell culture, *npr-17* cDNA was amplified from N2 cDNA using MyTaq DNA polymerase (Bioline, Taunton, MA) to generate PCR products with a 5′ Kozak consensus sequence and 3′-A overhang. This product was inserted into the PCDNA3.3 TOPO TA cloning vector (Invitrogen, Carlsbad, CA) according to the manufacturer's instructions.

Transgenic animals were generated by microinjection using a Zeiss Axio Observer A1 DIC microscope equipped with an Eppendorf Femtojet microinjection system. For the injection marker, we used *Punc-122::GFP* or *Punc-122::RFP*. 50 ng/µl of the fused construct or the plasmid was injected together with 50 ng/µl of the injection marker.

## Measurement of pharyngeal pumping rates in starved worms

Worms were grown on NGMSR plates seeded with *E. coli* HB101 until the first day of adulthood, then transferred to NGMSR plates without food, After 1 hr at 20°C, we measured the pumping rate by counting pharyngeal grinder movements for 1 min using a Wild M410 dissecting microscope at 64× magnification. For drug experiments, we prepared drug-containing plates by spreading 300 µl of a 10× aqueous drug solution on a 35 mm diameter plate, then waiting at least 3 hr for the drug to diffuse into the agar. First-day adults were then transferred to these plates and incubated for 1 hr at 20°C before counting.

## RNAi screen

RNAi was induced by feeding as described (*Timmons et al., 2001*), with the following modifications. Standard NGMSR agar was supplemented with 25 µg ml$^{-1}$ carbenicillin and 1 mM isopropyl β-D-1-thiogalactopyranoside and poured into 12-well plates. *E. coli* HT115 carrying the appropriate RNAi clones was grown in LB containing 100 µg ml$^{-1}$ carbenicillin at 37°C overnight, then 50 µl drops were seeded onto plates, making sure the culture dried within 1–2 hr, and induced at 37°C for 12 hr. Fourth stage larval (L4) worms were then transferred to the plates (3 worms per well) and allowed to grow to young adulthood at 20°C for approximately 4 days. We tested a total of 115 different neuropeptide genes. We used the Ahringer RNAi library for 91 neuropeptide clones (*Kamath and Ahringer, 2003*). For 24 neuropeptides, we cloned PCR-amplified exons into the L4440 vector. Primers for making these constructs are listed in *Supplementary file 3*.

## qRT-PCR

Total RNA was isolated using TRIzol (Life Technologies, Carlsbad, CA), and reverse transcribed using the Tetro cDNA Synthesis Kit (Bioline). The cDNA was quantified using Nanodrop (ND-1000) and used in qRT-PCR reactions. These reactions were performed on a CFX96 Real-Time PCR Detection System using the SensiMixPlus SYBR & Fluorescein Kit (Bioline). We used 500 ng of cDNA per sample in a total volume of 25 µl. Target genes were amplified using specific primers. Amplification and expression analysis were performed in triplicate. mRNA levels in the tested strains were normalized with *ama-1*, a housekeeping gene. For quantification, we used the Delta–delta Ct method. Primers were *nlp-3* Forward TGTGTCTACTCTGCTCCCTATG, *nlp-3* Backward TGATCATGTCTGGACGGAAAG for *nlp-3* and *nlp-24* Forward ACGGAGGTGGACGTTATGGA, *nlp-24* Backward GAGACCGCCTCCTCCGTAT for *nlp-24*.

## Cell culture and transfection

HEK-293 cells (ATCC, Manassas, VA) were cultured in DMEM supplemented with 10% FBS at 37°C in a humidified atmosphere containing 5% $CO_2$. For stable cell line generation, we transfected with a 1:1 ratio of pcDNA3.3 containing 1 µg *npr-17* and Gα15 cDNA (promiscuous G protein 15, Missouri S&T cDNA Resource Center) using X-tremeGENE HP DNA transfection reagent (Roche, Switzerland). Cells were seeded into 10 cm dishes and antibiotic (500 µg/ml G418) was added to the culture medium the next day. The selection medium was changed every 3 days until colonies formed. A single colony was picked, expanded, and tested by calcium imaging. Stable cell lines expressing human opioid receptor MOR-1, DOR-1 and KOR-1 were constructed similarly. MOR-1, DOR-1 and KOR-1 plasmids were provided by the Duke University GPCR Assay Bank.

## Calcium imaging

HEK-293 cells cotransfected with NPR-17, MOR-1, DOR-1 or KOR-1 receptors and Gα15 were plated onto 35 mm MatTek glass bottom dishes for calcium imaging. When they reached 70–80% confluency calcium imaging was conducted using the Fluo-4 Direct Calcium assay kit (Molecular Probes, Carlsbad, CA). The cells were loaded with Fluo-4 Direct Calcium reagent and incubated at 37°C for 50 min. For naloxone treatment, 10 µM naloxone was added in the Fluo-4 Direct Calcium reagent before incubation. Synthetic NLP-24 peptides (all synthesized with amidated C-termini), opioid agonists, and blockers diluted in HBSS (Hank's Balanced Salt Solution) buffer were applied to the cell for 10 s, and fluorescence at 530 nm was monitored (excitation wavelength 470 nm) with a Zeiss Axio Observer A1 microscope and ZEN software. Experiments were conducted on three plates for each condition on three different days. Fluorescence was quantified using ImageJ. We imported images using the ImageJ Bio-Formats plugin and subtracted background, then measured the mean Integrated Density.

## Locomotion assays

L4 worms were picked to an HB101-seeded NGMSR plate and given 12 hr to develop to adulthood. Adult worms were transferred to 6 cm NGMSR plates without food and incubated for 30 min. After 30 min, a single starved worm was then transferred to an individual 3.5 cm plate and its behavior recorded as described previously (*Gallagher et al., 2013b*). We used a 10 mm copper ring to constrain the worm. The light was then turned on and video capture proceeded at 1 frame/s for 1 hr. Recordings were made on a modified version of the nine-worm recording station described by *Shtonda and Avery (2006)* in which the worms were imaged through Computer MLM3X-MP macro zoom lenses onto Pointgrey GRAS-14S5M-C digital cameras. We used ImageJ to get $x$, $y$ coordinates. We inverted each image and subtracted background. The light/dark threshold was adjusted to find the outline of the worm then we determined the coordinates of the worm centroid using the 'Analyze Particles' command in ImageJ. Finally, we used HMM analysis to define locomotive behavioral states (*Gallagher et al., 2013a*).

## Statistical analysis

Pumping rates were tested for statistically significant effects by ANOVA and corrected for multiple testing by the Dunnett, Tukey, or sequential Bonferroni methods, as appropriate. Interactions (for instance, the difference between the effects of morphine and naloxone on wild-type and *npr-17* in *Figure 2*), were tested by two-way ANOVA. Gene expression (*Figure 2A*) was tested by the interaction between gene (*ama-1*, *nlp-3*, or *nlp-24*) and nutritional state (well-fed or starved), with well-fed *ama-1* as the base case. These tests were carried out on $C_t$ measurements, not gene expression levels, since gene expression measurements are typically heteroscedastic.

Calcium imaging results were more complicated. In each experiment, a series of measurements was made of the effects of different drugs or drug concentrations. All measurements in a series were related by the use of cells sampled from the same population. Each series also included a blank in which buffer was added. For each condition, we first measured base fluorescence ($f_0$), then added drug and measured fluorescence after 1 min ($f_1$). We then pooled all the series measuring a given effect and tested log ($f_1/f_0$) in a two-way ANOVA with series and drug as the factors, no interaction, and the blank as the base case. This effectively normalizes each measurement to the blank in its own series and tests the residues for drug effects. Particularly for *Figure 5B* and *Figure 6D* this test is conservative, since it makes no use of the normalization to the maximal effect measured with loperamide or U69593.

## Acknowledgements

This work was supported by research grant DK083593 from the US Public Health Service to LA. We thank Hamid Akbarali for providing morphine. *nlp-24* and *npr-17* strains were provided by the National Bioresource Project. Kristen Davis commented on the manuscript and helped with the locomotion assay.

## Additional information

### Funding

| Funder | Grant reference | Author |
|---|---|---|
| U.S. Public Health Service | DK083593 | Leon Avery |

The funder had no role in study design, data collection and interpretation, or the decision to submit the work for publication.

### Author contributions

MCC, Conception and design, Acquisition of data, Analysis and interpretation of data, Drafting or revising the article; ABA, Analysis and interpretation of data, Drafting or revising the article; Y-JY, Conception and design, Drafting or revising the article; LA, Conception and design, Analysis and interpretation of data, Drafting or revising the article

## Additional files

**Supplementary files**

• Supplementary file 1. Genes that affect the growth when knocked down by RNAi.

• Supplementary file 2. Primers for transgenic worms.

• Supplementary file 3. Primers for RNAi.

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
