## [Decision Letter]

Thank you for sending your work entitled “An opioid-like system regulating feeding behavior in *C. elegans*” for consideration at *eLife*. Your article has been favorably evaluated by a Senior editor and two reviewers, one of whom is a member of our Board of Reviewing Editors. The Reviewing editor and the other reviewer discussed their comments before we reached this decision, and the Reviewing editor has assembled the following comments to help you prepare a revised submission.

We think that this manuscript is of potentially significant interest and we would like to consider a revised version in which one key criticism needs to be addressed, which concerns the expression of *nlp-24*. The reporter analysis performed so far is insufficient and not interpretable due to the secreted nature of the reporter. We suggest to separate the *nlp-24* sequence from the reporter by either an SL2 sequence or a self-cleaving 2A peptide (as demonstrated in the literature). This should reveal the site of expression of *nlp-24* without the obscuring effect of secretion of the peptide-fused GFP. The *SL2::GFP* or *2A::GFP* reporter cassette should be inserted into the rescuing *nlp-24* genomic piece of DNA. The dependence of reporter gene expression on starvation should be tested.

It is also not clear why the authors do not discuss the previously reported expression of *nlp-24* in the ASI neurons (Nathoo et al.). This finding suggests that *nlp-24* may operate in an autocrine manner.

Other minor editorial comments:

1) In Figure 1 the authors should indicate significance the way they did in Figure 1: use lines to depict the comparisons made.

2) Some data that are in supplementary appears valuable enough to warrant inclusion of the data in the main figure: Figure 3—figure supplement 1 and Figure 3—figure supplement 2 and Figure 4–figure supplement 1, 2.

3) If existing, data should be added in Figure 6 on the following genotypes: *npr-17(-)*, *nlp-24* (overexpression) in wt background and in *npr-17* mutant.

---

## [Author Response]

*We think that this manuscript is of potentially significant interest and we would like to consider a revised version in which one key criticism needs to be addressed, which concerns the expression of* nlp-24*. The reporter analysis performed so far is insufficient and not interpretable due to the secreted nature of the reporter. We suggest to separate the* nlp-24 *sequence from the reporter by either an SL2 sequence or a self-cleaving 2A peptide (as demonstrated in the literature). This should reveal the site of expression of* nlp-24 *without the obscuring effect of secretion of the peptide-fused GFP. The* SL2::GFP *or* 2A::GFP *reporter cassette should be inserted into the rescuing* nlp-24 *genomic piece of DNA. The dependence of reporter gene expression on starvation should be tested*.

*It is also not clear why the authors do not discuss the previously reported expression of* nlp-24 *in the ASI neurons (Nathoo et al.). This finding suggests that* nlp-24 *may operate in an autocrine manner*.

Thank you for your suggestion. We tested *nlp-24::SL2::GFP* as you commented. As you expected we could see ASI expression in *nlp-24::SL2::GFP* transgenic worms (Figure 7—figure supplement 2). That suggest that *nlp-24* may operate in an autocrine manner. We also added a starvation experiment (Figure 7—figure supplement 3). *nlp-24* expression is increased by starvation. We didn’t see a big difference in *nlp-24* expression in ASI neurons. Therefore we think that nlp-24 may also operate in an endocrine manner. These new data Figure 7—figure supplement 2 and Figure 7—figure supplement 3 are now discussed in the Results section (in the subsection headed “ASI neurons are required for opioid signaling and NPR-17 in ASI neurons is sufficient for opioid signaling”) of the revised manuscript.

*Other minor editorial comments*:

*1) In*
Figure 1
*the authors should indicate significance the way they did in*
Figure 1*: use lines to depict the comparisons made*.

We did. We added lines in Figure 1.

*2) Some data that are in supplementary appears valuable enough to warrant inclusion of the data in the main figure:*
Figure 3—figure supplement 1 and Figure 3—figure supplement 2
*and Figure 4–figure supplement 1, 2.*

Figure 3—figure supplement 1 and Figure 3—figure supplement 2 became Figure 4. We made Figure 5 which included Figure 4 and Figure 4–figure supplement 1. And Figure 4–figure supplement 2 became Figure 9.

*3) If existing, data should be added in*
Figure 6
*on the following genotypes:* npr-17(-)*,* nlp-24 *(overexpression) in wt background and in* npr-17 *mutant.*

Unfortunately, we don’t have that data. We suspect *npr-17* will show similar feeding behavior to wild type. And we thought that wild type behavior was already highly activated by starvation, so I assumed that *nlp-24(O/E)* would also show similar behavior with wild type worms.